# Heart Rate Variability Biofeedback in Cancer Patients: A Scoping Review

**DOI:** 10.3390/bs12100389

**Published:** 2022-10-11

**Authors:** Gea Elena Spada, Marianna Masiero, Silvia Francesca Maria Pizzoli, Gabriella Pravettoni

**Affiliations:** 1Applied Research Division for Cognitive and Psychological Science, European Institute of Oncology, IRCCS, 20132 Milan, Italy; 2Department of Oncology and Hemato-Oncology, University of Milan, 20122 Milan, Italy

**Keywords:** cancer, biofeedback, heart rate variability, autonomic balance, decision-making, quality of life

## Abstract

Heart Rate Variability (HRV) Biofeedback (BFB) has been shown to improve autonomic balance and wellbeing in chronic diseases. As cardiac variability represents an index of cognitive and emotional regulation, HRV-BFB has been shown to lead to improvements in physiological and psychological adaptability and quality of life. However, knowledge of HRV-BFB in cancer patients is lacking, and available results are diversified according to methods and outcomes. The present paper undertakes a scoping review, exploring the use of HRV-BFB to modulate autonomic balance, cancer symptom management, and quality of life in cancer. This scoping review analyzes empirical evidence considering study designs, BFB methods, and psychophysiological outcomes. Research that focused on HRV-BFB effects in cancer patients was selected (79%). In addition, a systematic review and meta-analysis (31%) focusing on HRV, or BFB in chronic conditions, including cancer, were considered. The studies examined BFB treatment for thyroid, lung, brain or colon cancer, hematologic cancer, and survivors or terminal cancer patients. Retrieved studies reported physiological and psychological indices as primary outcomes: they included HRV values, sleep, pain, fatigue, depression, anxiety, and quality of life. Although the heterogeneity of publications makes it difficult to generalize the effectiveness of HRV-BFB, the training has been proven to improve cancer symptoms and well-being.

## 1. Introduction

Notwithstanding the scientific advancements achieved in cancer care, 19.3 million new cancer cases have been estimated, and 10 million cancer deaths in 2020 alone [1]. The global cancer burden is expected to be 28.4 million cases in 2040 [2,3]. Receiving an oncological diagnosis and dealing with treatments might be associated with psychological distress [3], depression, and anxiety [4]. However, as oncological prevention and health paths improve and treatments or therapeutic protocols evolve [5], more patients are living longer with a better quality of life (QoL) compared to the past [6]. This poses important challenges for the management of psychological and physical morbidities related to the disease and its treatments.

Biofeedback (BFB) is a technique that could be used to learn to control specific human body functions (i.e., heart rate, temperature, muscular tone, skin conductance, breathing). During BFB, the subject is connected to electrical sensors, that help people receive information about their bodies. The received feedback helps make subtle changes in the body (such as slowing down the heart rate or relaxing some muscles), achieving specific results (for example, reducing pain). These bio-signals provide audio-video feedback, and additional suggestions may be given by the biofeedback therapist if needed. Thanks to the received feedback, the patient increases his/her ability to change impaired bodily functions, with a positive effect on well-being, and experiences feelings of self-control, regulation, and efficacy [7]. Although other innovative treatments have been shown to improve relaxation and psychological symptoms in cancer conditions [8], evidence reports that BFB is an effective method that enhances self-competence by providing feedback on physiological signals from which cancer patients can benefit, increasing physiological self-regulation and psychological well-being [9,10,11,12]. Thus, BFB might be a valuable practice in cancer treatment, allowing better management of the psychological morbidities related to cancer, as well as pain and fatigue caused by the antitumor therapies, or other physical dysfunctions [13]. Such elements have been associated with altered homeostasis and autonomic balance [14]. Thus, treatments that focus on homeostasis and autonomic rehabilitation may show positive effects both on physiological and psychological cancer-related issues. HRV, which is often employed in BFB protocols, is a physiological phenomenon described as continuous fluctuations in the interval between consecutive heartbeats [normal-to-normal beats] [15]. HRV BFB protocols are developed according to experimental and clinical specific needs and may be heterogeneous in procedure and timeline. However, Leher et colleagues (2007; 2013) provide reliable guidelines for the rehabilitation of HRV index through biofeedback. The complex variability of heart rate is under the direct control of the central nervous system that receives and coordinates autonomic, endocrine, and behavioral responses. In line with this, HRV has been shown to be a valuable biomarker of general health, autonomic balance, adaptability and psychological well-being, both in healthy individuals and patients [16,17,18,19]. Autonomic balance is represented by an optimal interplay of sympathetic and parasympathetic influences on the sinus node, and it is associated with a higher survival rate and health status in both clinical and general populations [20]. Parasympathetic influences slow down the heart rate and are associated with increased vagal activity, digestion, resting, and social engagement. Instead of sympathetic activity fixes the beat and it is responsible for general mobilization [21]. The link between HRV and brain processes has been suggested in the neurovisceral integration model [22], which highlights the relationships between the autonomic nervous system and the attentional and affective systems through structural and functional networks, to explain cognitive and emotional regulation. In this context, HRV may be considered a key psychophysiological proxy, being an indicator of central-peripheral feedback mechanisms.

The relationship between the autonomic nervous system and cancer has been demonstrated in the literature [23,24]. For example, a study by Magnon and colleagues [25] showed that sympathetic influences were important at the first stages of tumorigenesis, while the parasympathetic fibers were affecting cancer progression at later metastatic phases of the disease. In line with this, another study showed a positive association between decreased HRV and severe pain in cancer patients [9]. Furthermore, research in the field of chronic pain assessed the relation between low HRV and psychological weakness (e.g., distress, cognitive or emotional dysregulation) [22,26]. Thus, several publications linked cardiac variability to health not only from a physiological perspective but also from a psychological point of view. In line with this, high HRV has been shown as a promising biomarker related to adaptability and well-being [27]. Specifically, in stressful conditions, HRV was associated with enhanced cognitive resilience, appropriate emotional regulation, and better modulation of cortisol, cardiovascular and inflammatory responses. To date, several studies have investigated the association between vagal nerve activity and the prediction of prognosis in cancer, highlighting the fact that HRV predicted patient survival [28,29]. Moreover, a systematic review asserted that valuable heterogeneous therapies for cancer patients (e.g., palliative care, relaxation therapies, music therapies, aerobic exercise, myofascial release sessions) had an effect on increasing HRV coherence or its absolute values [6]. HRV coherence is defined as a relatively harmonic (sine wave-like) signal with a very narrow and high-amplitude peak in the low-frequency domain of the power spectrum, with no major peaks in the other bands [30]. HRV coherence has been linked to favorable emotional states such as appreciation and compassion [31]. The increased HRV reported in the systematic review [6] is coherent with the positive outcome of selected therapies that would be associated with autonomic balance, health, and well-being, to reduce cancer-related fatigue and negative moods. However, given the limited amount of HRV data on cancer patients, the reported studies considered mixed cancer conditions; further studies focusing on specific cancer diagnoses are needed to validate the presented results.

### Aim

The current scoping review aimed to provide a descriptive overview of the effectiveness of HRV BFB in cancer treatment. Specifically, the impact of HRV BFB on the management of cancer symptoms was investigated, evaluating physiological (autonomic balance, HRV coherence, HRV values, pain, sleep disorders) and psychological (pain, fatigue, distress, depression, cognitive and emotional regulation) aspects. Furthermore, sub-aims considered current evidence on the relation between (1) autonomic balance and cancer prognosis and (2) HRV BFB and chronic pain.

## 2. Methods

Since the research on the HRV-BFB in the cancer domain is quite heterogeneous, the method of a scoping review was chosen, to provide a more comprehensive and descriptive picture in investigating the effectiveness of HRV BFB. The effectiveness of HRV BFB considers either physiological or psychological symptoms. Indeed, the scoping review methodology permits the building of a knowledge synthesis of the main concepts and theories in a given field, where evidence is ambiguous and heterogeneous, as in the field of BFB training. The present scoping review is conducted following PRISMA guidelines for scoping reviews [32,33] and Munn’s guidance for authors [34]. The five steps were followed: (1) identification of the research question, (2) identification of the relevant studies, (3) study selection, (4) data mapping, (5) comparison, summary, and reporting of results [35].

### 2.1. Inclusion and Exclusion Criteria

A set of inclusion and exclusion criteria were established to drive the studies selection. Studies were eligible for inclusion if they conformed to the following criteria: (a) research within the field of health (nursing, psychology, and psychiatry); (b) HRV assessment or BFB training in cancer patients; (c) studies in English, Italian, or Spanish languages; (d) research published from January 2008 (when the Biofeedback Alliance and Nomenclature Task Force defined biofeedback) to the present; (e) experimental and quasi-experimental; observational; qualitative; mixed-method; systematic reviews; meta-analyses; scope reviews; overview articles; and narrative reviews. Conversely, studies were excluded if: (a) pathologies other than chronic pain or cancer were treated; (b) physiological indices other than HRV assessment or training were administered to patients; (c) researchers did not report quantitative analyses.

### 2.2. Literature Search Strategies

The search databases were EMBASE, ProQuest Central, PsycINFO, PubMed, and Scopus. Sources of unpublished studies and gray literature such as congress abstracts, clinical trials and current controlled trials were included. An adapted search strategy was used for gray literature to avoid an extremely high number of irrelevant results. The combination of keywords, labels, and synonyms was as follows: cancer (or tumor, neoplasms), HRV (or heart rate variability, heart rate, autonomic balance, vagal tone), and BFB (or biofeedback). A manual search was conducted in the selected books and chapters [36]. The reference list of all relevant studies was screened for additional studies. After the removal of duplicates, articles were selected in a two-step process by two reviewers working independently at each step: in step 1, two reviewers (GES and MM) screened independently all the titles and abstracts returned for potential relevance. Each reviewer assessed the article against the inclusion and exclusion criteria. The proposed abstracts included for full-text review were compared by the two reviewers.

Step 2 was undertaken for each abstract selected. The full version of the article was retrieved. Two reviewers (GES and MM) ran independent full-text analyses. Reasons for the exclusion of full-text articles were noted by each reviewer and provided as an appendix in the full review. The proposed full-text articles for the review were compared by the two reviewers until a final set was agreed upon by both. Where there was a disagreement between the two reviewers in either step, the final agreement was sought by mutual consensus with input from a third reviewer (SFMP). Please see Table A2 in Appendix B for further details.

### 2.3. Data Charting and Summarizing Data

Two reviewers (GES and MM) independently extracted study data. Data to be extracted from selected studies included information about the author(s), year and country of the study, study design, goals, sample characteristics, guidelines of intervention, BFB methodologies, and any other key findings related to the research questions. The methodology and statistical quality of the studies were not discriminative in the presented scoping review because the goals of this study were to identify gaps in the literature and to propose potential research questions for future systematic review [35].

## 3. Results

### 3.1. Selection of Sources of Evidence

A first search in the literature identified 6998 potential articles. The search string included the following words: (((“Heart Rate Variability *”) OR (“HRV*”) OR (“Autonomic*”) OR (“Vagal Tone*”) OR (“Heart Rate”)) AND ((“Cancer*”) OR (“Tumor*”) OR (“Carcinoma*”)) AND (“Biofeedback*”)).

After all duplicates and inappropriate titles and abstracts were removed, twenty-two full-text articles, a book chapter, and five abstracts were screened (by GES), in consultation with the second author (MM), for eligibility. Overall, nine studies that did not meet the inclusion criteria were excluded from the scoping review. After a three-part screening process, nineteen studies were considered. Specifically, eleven original research studies, four systematic reviews and meta-analyses, and four abstracts were included. Original research might be included or cited in the systematic reviews and meta-analysis publications. The majority of the studies (34%) were randomized control trials or cohort studies (40%) and there were also single case studies (13%). Finally, a non-experimental and a pilot study (13%) were considered. A total of 82% of included studies were conducted with adults, with 18% on children and adolescents. Selected publications (79%) included primary and secondary data.

The diagram of the selection of sources process is presented in Figure 1. Relevant data for each included source of evidence are reported in Table 1 (empirical studies) and Table 2 (theoretical studies).

### 3.2. Synthesis of Results

Overall, the empirical and theoretical studies included in this review highlighted the potential effectiveness of HRV BFB interventions in cancer care. The BFB method was used alone or in combination with other treatments; included studies reported HRV BFB in association with (a) biofeedback of other physiological signals [37]; (b) physical exercise [38]; (c) resonant, paced, and belly breathing exercises [39,40,41]; (d) breakthrough pain treatments [42]. Further, considered studies investigated single assessments or full trainings on a daily–weekly basis with different duration, ranging from 10 s assessments [43] to 24 h measurements [42]. However, the most common duration for BFB trainings was between 5 and 60 min [40,44,45]. Most interventions [39,40,44,45,46,47,48,49] implicated hospital management of the treatment and the support of a BFB therapist, while others consisted of at-home self-training [41], and a minority were a mixed approach [50]. Specifically, home trainings were developed by implementing respiration and relaxation exercises on a deliverable device that monitored cardiac variability and obtained physiological data without the physical presence of patients.

The included interventions considered both single case studies [39] and large sample researches based on up to 272 cancer patients [45]. Selected studies targeted adult [38,41,42,45,46,47,48,51,52,53,54], adolescent [49], and child [39,40] cancer patients. According to the given information, although BFB training was extremely heterogeneous, there were no specific differences in BFB trainings based on the mean age of the included sample.

Explored outcomes reported pain, fatigue, sleep quality, depression, anxiety, HRV coherence, frequency HRV indexes, stress, and quality of life. Overall, a positive effect of HRV BFB on all these variables was observed. Specifically, cancer pain rates were lowered with the increase of HRV and according to self-report scoring [42,50,53], and so were fatigue scores [53]. Increased sleep efficiency, sleep duration and a general reduction of sleep symptoms, insomnia, and daytime impairments were registered after paced breathing and BFB training [41,47,50,53]. The levels of depression and anxiety were compared before and after HRV BFB trainings, reporting a significant reduction [48,53]. However, not all reported publications reached significant improvements in psychological aspects such as quality of life (QoL) and distress e.g., [47,49] Finally, HRV measurements and coherence scores (i.e., a measure of the degree of coherence in the heart rhythm pattern) were increased, following the BFB and respiration exercises in several studies [38,53], mirroring a better autonomic balance than in the pre-treatment condition. A synthesis of the most common outcomes extracted from the selected publications is reported in Appendix A.

## 4. Discussion

The current scoping review explored the use of HRV BFB in different types of cancer, exploring the impact both on physiological, physical, and psychological wellbeing. As previously reported, frequent cancer symptoms may include severe pain, fatigue, sleep and eating disorders, distress, depression, anxiety, and general psychological burden. Therefore, this population may particularly benefit from HRV BFB treatment [36]. Furthermore, a study conducted by De Couck and colleagues showed that HRV emerged as a potential variable inversely related to tumoral biomarkers, and that a low HRV was measured in cancer patients compared with healthy controls [55]. Evidence analyzed in the current scoping review suggested the potential effectiveness of HRV BFB on the management of physiological variables (HRV coherence and values to evaluate autonomic balance, sleep disorders), and psychological symptoms (depression, anxiety, distress, quality of life); pain and fatigue in cancer patients were explored, demonstrating a general positive effect of this treatment within the cancer population. The impact of HRV BFB was measured on primary and secondary outcomes of the included studies. Overall, included studies were conducted on thyroid cancer, lung cancer, brain cancer, colon cancer, hematologic patients, and in cancer survivors or terminal patients.

In most cases, the studies analyzed highlighted the fact that HRV BFB showed positive results concerning pain levels, depression and anxiety, sleep disturbances and cognitive performance. Three randomized controlled studies [41,46,53] showed a significant increase in HRV values and improvements in sleep quality, fatigue, stress, and depression in patients that received BFB training, compared with controls. However, the BFB treatments markedly changed depending on training time, hole duration, and modalities between the studies, and the results should be interpreted cautiously. Similarly, Burch and colleagues (2020) found the greatest effect of the HRV BFB training on sleep symptoms and sleep-related daytime impairments. However, minor results in fatigue and distress scores were reported in this investigation. Furthermore, these results confirmed that increased HRV may facilitate homeostasis and cardiorespiratory synchronization leading to better sleep quality [55]. Some retrieved studies have investigated the procedure used to deliver the BFB at home [38,41,47] and in hospital settings [37,39,40,44,46,47,52,53]. For example, Hasuo and collaborators (2020) tested a program to improve sleep by implementing a home-based HRV BFB with a deliverable device. This approach led to significant results in sleep function and HRV, with patients reporting improvements in sleep induction disorder, nocturnal awakening and unrefreshing sleep, in only two weeks. The use of home-based BFB allowed patients to exercise daily and acquire technical respiration skills in a shortened period, compared with hospital interventions. This is in line with the active role in controlling lifestyle and decisions that patients should embrace [56,57]. Studies in which the BFB intervention was performed in hospital reported high rates of dropout and reduced feasibility of completing the purposed treatment [37]. Ozier and colleagues (2018) adopted a mixed method associated with hospital-training home exercises, and their method showed good patient adherence to the intervention. Generally, we argue that given the fact that cancer patients may particularly benefit from treatments that enhance QoL in a limited time, this could be a major advantage to home training or a mixed approach.

Selected cohort and case series studies reported improvement in physiological outcomes, depression, and anxiety [38,39,40,44,47]. In one of these [38], BFB was associated with physical exercise, and it brought improvements in physical capacity and muscle strength in association with a decrease in fatigue rates, although this last result did not reach statistical significance. As regards the psychological symptoms, results confirmed the effect of augmented cardiac variability on the reduction of depression and anxiety in the clinical population [16]. In the main, cancer patients who underwent HRV BFB intervention had lower scores for depression and dysphoric mood [47,53], and distress and anxiety levels [37,53].

Overall, the majority of the selected publications reported either physiological and/or psychological improvements: positive results were registered independently of the type and stage of cancer and were a confirmation of the possible beneficial effect of HRV BFB on psychophysiological adaptability. This is in line with the idea of HRV BFB as a possible therapy alone or combined with other treatments in the complex health path of cancer. The current scoping review also analyzed a set of theoretical studies on HRV suggesting it as a possible factor associated with cancer survival rate [43,58,59]. DeCouck and colleagues [43] reported the beneficial effects of vagal nerve stimulation on survival rate, measured in a noninvasive manner through the HRV index. Similarly, another group of studies highlighted the fact that that low vagus nerve activity was related to poor health outcomes, as well as vagus nerve stimulation slowing down tumor progression [60,61,62,63]. Furthermore, these studies showed that HRV significantly predicted tumor marker levels in different cancers, and that higher vagal activity predicted a better prognosis of cancer conditions. Zhou and colleagues [34], in a systematic review and meta-analysis pooled together the results of six studies with HRV measurements, demonstrating that survival was significantly longer in the higher HRV group than in the lower HRV group. However, the authors recommended caution in the interpretation of obtained results, given the common problem of heterogeneity of methods (HRV assessment guidelines and instruments) and samples (pancreatic cancer, breast cancer, lung cancer, and mixed cancer type, all metastatic). Overall, we might affirm that the considered theoretical studies report show links between cancer and low vagal influence and HRV [43,59] and they observe a positive association between high HRV and survivorship [58]. BFB represents a way of increasing cardiac variability. In line with the preliminary HRV BFB effectiveness on physiological and psychological outcomes shown by the reported publications, more attention should be given to HRV BFB, as cancer health paths could benefit from the treatment.

### 4.1. Implications and Suggestions for Research and Practice

The presented scoping review pointed out the need for additional original studies on HRV BFB to validate the presented results. Shared treatment standards (i.e., frequency, duration, and follow-ups) should be implemented in future investigations to adequately compare outcomes. Moreover, there is an evident need for additional studies to achieve empirical data on the feasibility and outcomes of HRV BFB, according to specific criteria such as type stage, the prognosis of cancer, age of patients, and previous or current presence of other treatments. Better statistical designs compared with currently available investigations are suggested, for example, larger sample sizes and dropout prevision should be implemented, to avoid inconsistent results.

Based on currently available data, it seems that HRV BFB showed major effectiveness when hospital-based treatments were turned into home-based training through deliverable devices or at least associated with self-practice. Indeed, these modalities allowed patients to establish a time to achieve results, and prevented the potential issue of dropout. More studies embracing these BFB deliverable methodologies targeting specific cancer populations are encouraged.

### 4.2. Limitations

A rigorous approach to extract, search and evaluate the existing literature was adopted. However, the present scoping review had some limitations that should be considered. Firstly, retrieved studies showed a divergence of methods in physiological recordings, BFB instruments, BFB training (e.g., hospital and/or home-based), and follow-up evaluations. Secondly, given the inability to completely fulfill the research questions because of the lack of available sources, miscellaneous samples of cancer patients were included: mixed samples considered all cancer types (e.g., breast cancer, lung cancer, pancreatic cancer, mixed cancer), stages (I, II, III, IV) age ranges (from children to adulthood) and gender. In addition, included studies contained a lack of assessed control variables (e.g., physical activity, smoking habits, etc.), medium-to-high dropout rates, and missing data. Thirdly, the gray literature was included in the current scoping review to draft a better picture of the accruing evidence on HRV-BFB.

Differentiated studies according to cancer type and stage, and age and gender of patients, as well as larger sample sizes, should be encouraged for future studies. Similarly, common guidelines and standardized protocols in HRV measurements and BFB training should be available and followed by researchers and clinicians.

## 5. Conclusions

Evidence collected in the current scoping review revealed that HRV BFB might help to increase HRV coherence and values, thus improving both physiological symptoms (e.g., autonomic balance, general health condition, and sleep quality), psychological symptoms (e.g., fatigue, depression, anxiety) and pain, in cancer patients. Specifically, retrieved results showed an increase in autonomic balance, thus an increased HRV and/or HRV coherence, linked to physiological health and wellbeing. In addition, sleep quality and quantity seem to benefit from home-based HRV BFB training. Finally, selected studies showed promising results on reported fatigue, depressive and anxiety scores after treatment, compared with the previous condition. No differences between different diagnoses or grades of cancer were explored, given the need for further studies to explore the issue. In line with current results, clinicians might consider including HRV BFB either as home intervention or hospital-based training in the health path of oncological patients. Overall, this type of technique might be a challenging opportunity to better manage not only physical morbidities which are cancer-related but also psychological distress, improving health-related QoL along the cancer pathway. However, further high-quality studies are needed to establish reliable standards, as the heterogeneity of selected publications concerning method and sample makes it difficult to generalize results and assess the effectiveness of HRV BFB intervention.

## Figures and Tables

**Figure 1 behavsci-12-00389-f001:**
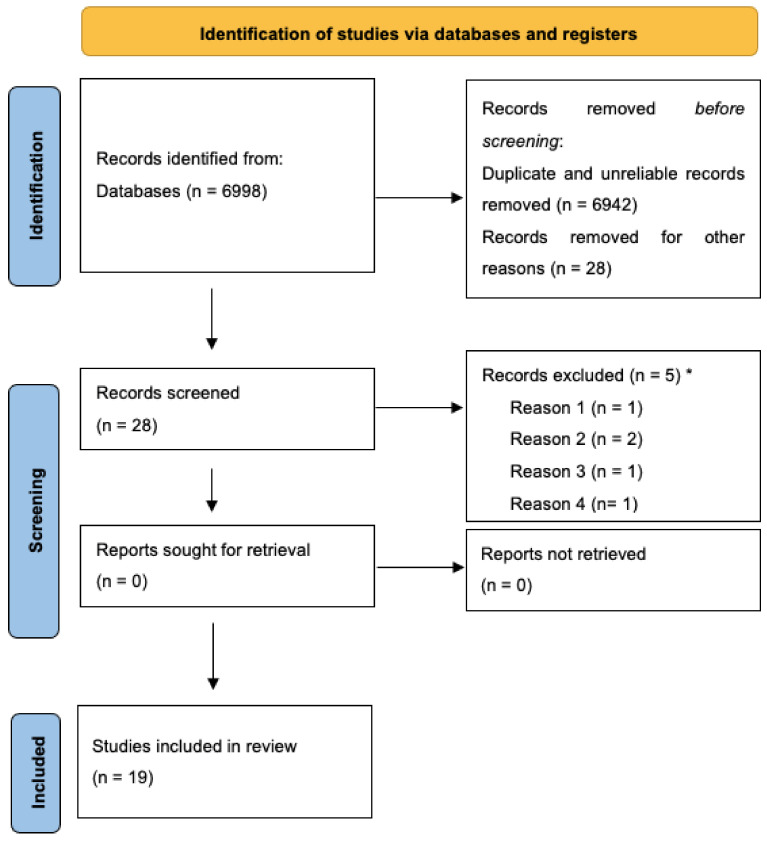
PRISMA 2020 flow diagram for the review process. * Reason 1: BFB on other physiological signals (e.g., electromyography, temperature, skin conductance); reason 2: HRV assessment without BFB; reason 3: unclear primary or secondary outcomes; reason 4: sample of patients without cancer diagnosis. Reprinted with permission from Page et al. (2021). Copyright 2020 The PRISMA statement.

**Table 1 behavsci-12-00389-t001:** Empirical original research studies characteristics and results.

Authors	Intervention and Study Design	Study Objectives	Sample Characteristics and Cancer Type	Biofeedback	Treatment Procedure	Outcomes
Greenberg et al., 2015	BFB multi signal assessment; feasibility study	Feasibility of HRV BFB and respiration training and improvement in QoL, disease-related symptoms (pain, stress, breathlessness and emotional distress)	8 participants; mean age 68 years (SD = 9.5); 4 women and 4 men; non-small cell lung cancer	Thought technology ProComp Infiniti 8-channel, Bio- Graph Infiniti (5.1.1.)	Electromyography; skin conductance; temperature; HR; respiration rate recordings; 6 training sessions (30–45 min)	Improvement in participants’ adaptation to cancer; mild levels of anxiety and depression, moderate distress, and somewhat reduced QoL; no variation in physiological recordings, despite a reduction in respiration and HR
Hasuo et al., 2020	Daily home BFB HRV and resonant breathing; experimental randomized comparative study	Develop self-coping through sessions of home-based HRV BFB with resonant breathing in patients with sleep disturbances; examine its short-term efficacy and feasibility	25 patients; 25 HC; mean age 66.4 years (SD = 12.5); 20 women and 30 men; miscellaneous cancer types	Portable HRV-BFB device with resonant breathing	Resonant breathing 5–7 days training (>5 min prior to bedtime)	Improvement in sleep efficiency, sleep duration and low-frequency component of HRV
Fournié et al., 2022	Physical exercise and daily home BFB HRV; feasibility study	Examine the feasibility of a program with BFB training; Symptom improvement (fatigue, physical function, HRV)	17 patients; mean age 54.5 years (SD = 11.7); 8 women 9 men; hematologic cancer	Symbiofi cardiac coherence software (SymbioCenter technology) and DODOW, LIVLA technology	12-week rehabilitation program including 24 supervised sessions of physical exercise associated with 10 supervised sessions of HRV BFB (1 h training; specific rate of 6 breaths/min) and daily home-based practice of paced breathing (20 min)	Improvements in physical capacity; muscle strength; and flexibility; coherence ratio and low-frequency spectral density of HRV signal increased. No changes in fatigue and static balance
Burch et al., 2020	BFB HRV; experimental randomized controlled trial	Evaluate the effect of HRV BFB on improving HRV coherence and reducing cancer-related symptoms	17 patients; control group 17 patients; mean age 60 years (SD = 3); 29 women 5 men; miscellaneous cancer types	EmWave pro system; portable emWave2	4–6 weekly training sessions (25 min coaching + 15 min personal practice) and home practice	Increase in HRV coherence ratio, decrease in sleep symptoms and daytime impairment scores, minor results in fatigue and stress
Murguìà et al., 2017	BFB HRV and paced breathing; single case study	Effectiveness of HRV BFB on self-regulation during radiotherapy	1 patient (female) 11 years old; ewing sarcoma	EmWave system	28 training sessions received during radiotherapy with diaphragmatic breathing	Relaxation, cognitive and emotional regulation
Groff et al., 2010	BFB HRV; case series study	improved HRV coherence and QoL	6 patients; age range 42–66 years (SD = 7.18); all women; breast cancer survivors	HeartMath BFB	17–23 sessions (30 min); 3 phases (*JustBe, Manage Stress, techniques*)	improved HRV scores in “*JustBe*” and “*Manage Stress*” BFB phases;
Shockey et al., 2013	BFB HRV and belly breathing; feasibility study	Testing a combined treatment that merged relaxation techniques and BFB as a novel coping strategy for procedural distress	12 patients; mean age 11 years (SD = 2.08); 5 female 7 male; miscellaneous cancer types	HeartMath EmWave BFB system	4-session intervention combining relaxation and BFB (60 min)	Upward trend of HRV coherence in each session; the combination of belly breathing and BFB techniques allowed the participants to feel in charge of their bodies before their procedures
Gidron et al., 2019	HRV, CA19–9 and NFL biomarkers assessment; retrospective observational study	Exploring associations between neurophysiological vagal nerve index and peripheral disease biomarkers predict prognosis	Sample 1 included 272 patients with advanced PC mean age 60 years (SD = 11.5); 48% female; sample 2 included 118 patients with MS; mean age 46.7 years (SD = 9.2); 64% female pancreatic cancer and multiple sclerosis	Electrocardiograms (10 sec–5 min)	n.a.	Cancer patients: HRV inversely related to the tumor marker CA19-9 in patients who later survived but not in those who died; MS patients: HRV was inversely related to NFL only in those who did not relapse
Hunakova et al., 2018	HRV, SpO2 and tcpO2 assessment; pilot study	Determine Tcpo2 and balance ANS by linear and non-linear HRV analysis in patients ongoing therapy, in remission and in HC	15 patients (breast tumor and remission) and 7 HC; patients mean age 55 years (sd = 14.58); controls mean age 52 years (SD = 13.25); all women; breast cancer patients and survivors, HC	2-modules précise 8008, EmWave Pro (5 min); tcpO2 and SpO2 recordings; HRV recordings	n.a.	Decrease in TcpO2 only in patients with current breast cancer and ongoing therapy. HR control and cardio-vagal regulation impaired in breast cancer patients compared with HC
Masel et al., 2016	HRV assessment and breakthrough pain treatment; cohort pilot study	Changes in the ANS were studied by measuring HRV during opioid therapy for cancer breakthrough pain in patients and comparing these changes with the reported pain levels	10 patients; mean age 62 years (SD = 5.2); 3 men, 4 women; advanced cancer patients	Portable 5-point electrocardiogram; 24-h peak-to-peak HRV measurement	n.a.	Positive correlation was found between opioid-induced reduction in patient-reported pain intensity, based on NRS and changes in LF/HF
Cosentino et al., 2018	HRV assessment; observational study	Evaluate the psychological and physiological adjustment in survivors	38 patients; age range 29–80 years, all women; ovarian cancer survivors	EmWave Pro HeartMath; basal HRV recording (6 min)	n.a.	High perceived general social support; general concern about appearance; good total QoL with specific difficulties in emotional functioning. HRV values were lower when compared to normative values
Ozier et al., 2018	BFB HRV; feasibility cohort study (abstract)	Test the feasibility of HRV BFB with distressed cancer survivors	9 patients; primary brain tumor (grade II–IV)	Procomp 2 (Thought technology)	HRV BFB task (10 Min); 8 weekly meetings; daily homework involved 20 min of paced breathing	HRV BFB completers’ posttraining NLF mean was larger than the pretraining mean. Decrease in Beck Depression Inventory-II (BDI-II) and Beck Anxiety Inventory (BAI)
O’Rourke et al., 2017	BFB HRV; experimental randomized controlled trial (abstract)	Improvement in HRV coherence, autonomic health, reduce symptoms (insomnia, pain, fatigue, depression, and stress)	31 patients, cancer survivors; miscellaneous cancer types	n.a.	Weekly HRV BFB training up to six weeks	BFB for cancer survivors improves HRV coherence and reduces insomnia, pain, fatigue, depression, and stress
Chen et al., 2017	BFB HRV; non-experimental design (abstract)	Evaluate the effect of the game-based BFB for respiratory training HRV, stress, and resilience in cancer	20 adolescent patients; age range 12–20 years; miscellaneous cancer types	ProComp Infiniti BFB	HRV BFB with dramatic games	Effect of HRV BFB: (1) no significant differences in perceived stress and resilience were shown between two groups, (2) gender differences in LF and SDNN (female lower), (3) group difference in PNN50, (4) correlates of resilience were gender and stress
Gidron et al., 2017	BFB HRV; experimental matched control (abstract) pilot study	Effects of vagal activation via HRV BFB on levels of a tumor marker CEA	6 patients; colon cancer (grade IV)	n.a.	20 min/day, for 3 months	Levels of CEA declined in the HRV BFB condition, a difference approaching statistical significance at 3 months. Patients were able to perform HRV BFB self-intervention daily and it appeared feasible.

Note: HRV = Heart Rate Variability; BFB = Biofeedback; HR = Heart Rate; QoL = Quality of Life; SD = Standard Deviation; HC = Health Controls; NFL = Neurofilament Light Chain; PC = Pancreatic Cancer; MS = Multiple Sclerosis; SpO2 = Blood Oxygen Saturation; tcpO2 = Trans-cutaneous O2 Partial Pressure; N.a. = Non-Applicable; ANS = Autonomic Nervous System; SDNN = Standard Deviation Normal to Normal NN (R-R) intervals; LF = Low Frequency; LF/HF = Low Frequency/High Frequency Ratio; PNN50 = the proportion of NN50 divided by the total number of NN (R-R) intervals.

**Table 2 behavsci-12-00389-t002:** Theoretical Studies Characteristics and Results.

Authors	Intervention and Study Design	Study Objectives	Sample Characteristics and Cancer Type	Biofeedback	Treatment Procedure	Outcomes
Fournié et al., 2021	Systematic review	Exploring HRV BFB effectiveness	1127 patients (sample size 10–210 patients); chronic pain disorders (including cancer)	Miscellaneous BFB instruments	Miscellaneous BFB trainings.	Feasibility of HRV BFB in chronic patients without adverse effects. Significant positive effects were found in patient profiles for hypertension and cardiovascular prognosis, depression and anxiety, sleep disturbances, cognitive performance and pain, which could be associated with improved QoL.
De Couck et al., 2018	Systematic and comprehensive review	Vagal influences in cancer prognosis	1822 patients; miscellaneous cancer types	Miscellaneous BFB instruments	Miscellaneous BFB assessment	HRV has prognostic value in cancer, predicting both survival and tumor markers in several cancers.
Zhou et al., 2016	Systematic review and meta-analysis	Predictive value of HRV in the survival of patients with cancer	1286 patients; miscellaneous cancer types	Miscellaneous BFB instruments	Miscellaneous BFB assessment	Overall survival was significantly longer in the higher HRV group than in the lower HRV group
Arab et al., 2016	Systematic review	HRV as an indicator of independent risk factor for health impairments	840 patients; breast cancer and breast cancer survivors	Miscellaneous BFB instruments	Miscellaneous BFB assessment	Clinical importance of autonomic modulation in patients and survivors is demonstrated by association with effects of surgery and treatments, and the adverse effects of surgery and treatments on survivors (cardiotoxicity, fatigue, and stress)

Note: HRV = Heart Rate Variability; BFB = Biofeedback; N.a. = not-Applicable; QoL = Quality of Life.

## Data Availability

Not applicable.

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
