# Peer review of "Heart Rate Variability Biofeedback in Cancer Patients: A Scoping Review"

_behavsci, 2022, doi:10.3390/bs12100389_

Round 1

Reviewer 1 Report

This article reviews published results obtained by various research teams on the response of heart rate variability (HRV) to specific stimulation of the vagus nerve called biofeedback (BFB) in patients suffering from various cancers. The results of latest work (15 articles) in this area have been standardized to  consistently present them in the table I. The results of earlier same type reviews ( 4 reviews)  are presented in table II.

The article could be of interest to a wide group of oncologists and clinicians dealing with the well-being of patients in oncology therapy  if the presentation was clearer.  To improve readability of the article

1) the protocol of  BFB to  HRV treatment is needed

2) the measures of HRV used in various examinations ( especially the HRV coherence)  should be defined

3) the coherence score needs a definition 

In addition, the tables should be modified,  so that the fields of the record can be read in lines rather, than divided into multiple lines of narrow columns.

Author Response

We are sending the revised version of our manuscript titled " Heart Rate Variability Biofeedback in Cancer Patients: A Scoping Review" to The Journal of Behavioral Sciences.

We thank you for the opportunity and your encouraging comments on our work. We have worked through the paper, incorporating suggestions. Modifications in the manuscript have been highlighted in red. Following you can find a response for each suggestion received and a reminder to the main document. Please feel free to contact us if you have any further questions regarding our manuscript.

We wish to thank you again for your consideration.

Consulting Reviewer 1: R1

R1: This article reviews published results obtained by various research teams on the response of heart rate variability (HRV) to specific stimulation of the vagus nerve called biofeedback (BFB) in patients suffering from various cancers. The results of latest work (15 articles) in this area have been standardized to consistently present them in the table I. The results of earlier same type reviews (4 reviews) are presented in table II.The article could be of interest to a wide group of oncologists and clinicians dealing with the well-being of patients in oncology therapy if the presentation was clearer.  To improve readability of the article

  • the protocol of BFB to HRV treatment is needed

We thank the reviewer for this feedback. We added references to credited guidelines for HRV BFB protocol in the introduction section. Please see lines 62-63. We added a detailed procedure of the BFB trainings for all the included studies in Table 1. However, the description of the training was missing in some papers. We recognize that one of the major limitations of the current study is the heterogeneity of BFB methodology and the lack of shared protocols of the scientific literature. We explicitly addressed this limitation in the last part of the manuscript. Please see Limitations section 4.2.

  • the measures of HRV used in various examinations (especially the HRV coherence) should be defined

We added the legenda for all missing HRV indexes whether missing. Please see notes below Table 1 lines 233-236.

  • the coherence score needs a definition 

We added the definition for coherence HRV in the introduction section. Please see lines 97-100.

  • In addition, the tables should be modified, so that the fields of the record can be read in lines rather, than divided into multiple lines of narrow columns.

We appreciate the feedback and agreed with the need of revising the structure of the tables which have been revised according to the reviewer suggestion. Please see pages 7-11.

Reviewer 2 Report

The objective of this article is “a scoping review exploring the use of HRV-BFB to modulate autonomic balance, cancer symptoms management and quality of life in cancer.” Heart rate variability biofeedback (HRV-BFB) studies, which have focused on HRV (heart rate variability) as “one of the most important indicators of adaptive processes in the human organism” [Krivonogova, O. et al. (2021)], indicate the importance this has on neural oscillations, or brainwaves, may include many still-on-exploration role such as feature binding, information transfer mechanisms and the generation of rhythmic motor output on pain levels, depression and anxiety, sleep disturbances and cognitive performance [1].

Then, the authors address themselves to inquiry an overview about HRV BFB as a promising biomarker related to adaptability and well-being, especially in relation to cancer prognosis. See lines 99 forwards. Thalamocortical oscillation is associated with the appearance of specific mental states depending on the frequency range of the most prominent oscillatory activity, gamma most associated with conscious, selective concentration on tasks, learning (perceptual and associative), and short-term memory [2]. All these features may constitute the ground over which detect “physiological (autonomic balance, HRV coherence, HRV values, pain, sleep disorders) and psychological symptoms (pain, fatigue, distress, depression, cognitive and emotional regulation) in cancer patients.”

This scoping review is collected at Section 2. Methods. Inclusion and exclusion criteria are either temporal (since 2008 until now) and methodological. See lines 129-141.

Section 3. Results, explains, through Figure 1. at line 182, how the scoping review process has been conducted. In general, it is noticed in these reviews’ patients’ cancer pain rates as well as fatigue scores were lowered about the increase of HRV [3][4]. See lines 205-210.

The authors conclude their paper highlighting a positive effect of HRV BFB on many variables connected with cancer prognosis was observed. See lines 345-352.

Author Response

We are sending the revised version of our manuscript titled " Heart Rate Variability Biofeedback in Cancer Patients: A Scoping Review" to The Journal of Behavioral Sciences.

We thank you for the opportunity and your encouraging comments on our work. We have worked through the paper, incorporating suggestions. Modifications in the manuscript have been highlighted in red. Following you can find a response for each suggestion received and a reminder to the main document. Please feel free to contact us if you have any further questions regarding our manuscript.

We wish to thank you again for your consideration.

Consulting Reviewer 2: R2 

R2: The objective of this article is “a scoping review exploring the use of HRV-BFB to modulate autonomic balance, cancer symptoms management and quality of life in cancer.” Heart rate variability biofeedback (HRV-BFB) studies, which have focused on HRV (heart rate variability) as “one of the most important indicators of adaptive processes in the human organism” [Krivonogova, O. et al. (2021)], indicate the importance this has on neural oscillations, or brainwaves, may include many still-on-exploration role such as feature binding, information transfer mechanisms and the generation of rhythmic motor output on pain levels, depression and anxiety, sleep disturbances and cognitive performance [1].Then, the authors address themselves to inquiry an overview about HRV BFB as a promising biomarker related to adaptability and well-being, especially in relation to cancer prognosis. See lines 99 forwards. Thalamocortical oscillation is associated with the appearance of specific mental states depending on the frequency range of the most prominent oscillatory activity, gamma most associated with conscious, selective concentration on tasks, learning (perceptual and associative), and short-term memory [2]. All these features may constitute the ground over which detect “physiological (autonomic balance, HRV coherence, HRV values, pain, sleep disorders) and psychological symptoms (pain, fatigue, distress, depression, cognitive and emotional regulation) in cancer patients.” This scoping review is collected at Section 2. Methods. Inclusion and exclusion criteria are either temporal (since 2008 until now) and methodological. See lines 129-141. Section 3. Results, explains, through Figure 1. at line 182, how the scoping review process has been conducted. In general, it is noticed in these reviews’ patients’ cancer pain rates as well as fatigue scores were lowered about the increase of HRV [3][4]. See lines 205-210.The authors conclude their paper highlighting a positive effect of HRV BFB on many variables connected with cancer prognosis was observed. See lines 345-352.

Thank you for the comments. We read the comments carefully and checked the relevant literature highlighted by the Reviewer.

  • In relation to Table 1. at line 220 I would proceed the following:

I would create a little case glossary with information on the most recurrent topics explored in the reviews you have collected in relation to HRV-BFB (Heart Rate Variability - Biofeedback):

Then, I would create a simple Table with percentage and/or rate of occurrences of these topics in the reviews you have collected. You could further detail the Table per “cancer type” and/or “treatment procedure”.

We implemented all the suggestions of the reviewer to make contents more evident. We added a short glossary with the main terms used in the review and we added a new table with percentage rate of primary and secondary outcomes in Appendix A section.

  • In relation to Table 2. at line 231 I sincerely do not see the need to have “Study design” as one of the items, cos they are more or less all “systematic reviews”. Reducing the items you could advantage the distribution of text, that is less appealing (beyond the information you certainly give to the audience) when is long and disorderly distributed like yours.

We agreed with the request of the author to make tables format and content more appealing and understandable. Please see pages 7-11.

  • Please number sub-paragraphs. 

We thank the reviewer for this feedback. We numbered the sub-paragraphs to make the article structure.

Reviewer 3 Report

The paper is interesting and has the potential to publish in the above journal after the authors will address the following major comments:

- The abstract and the introduction are well written. 

- The following papers can be relevant to the considered research: 

1: Nave, O. (2022). A mathematical model for treatment using chemo-immunotherapy. In Heliyon (Vol. 8, Issue 4, p. e09288). Elsevier BV. https://doi.org/10.1016/j.heliyon.2022.e09288

2: Cho, H.-M., Kim, H., Yoo, R., Kim, G., & Kye, B.-H. (2021). Effect of Biofeedback Therapy during Temporary Stoma Period in Rectal Cancer Patients: A Prospective Randomized Trial. In Journal of Clinical Medicine (Vol. 10, Issue 21, p. 5172). MDPI AG. https://doi.org/10.3390/jcm10215172

- The authors describe the method and immediately after that the result of the paper. The method is not clear. What is the aim of the research? how do the authors achieve their aim? How used the method described?

- The authors used big data? If yes. Why they didn't use machine learning?

- The tables present in the paper are not clear. The authors should explain in more detail the present results.

- The conclusion section must be extended. Such that it should include the main results of the paper. The application of the results etc. 

Author Response

We are sending the revised version of our manuscript titled " Heart Rate Variability Biofeedback in Cancer Patients: A Scoping Review" to The Journal of Behavioral Sciences.

We thank you for the opportunity and your encouraging comments on our work. We have worked through the paper, incorporating suggestions. Modifications in the manuscript have been highlighted in red. Following you can find a response for each suggestion received and a reminder to the main document. Please feel free to contact us if you have any further questions regarding our manuscript.

We wish to thank you again for your consideration.

Consulting Reviewer 3: R3

R3: The paper is interesting and has the potential to publish in the above journal after the authors will address the following major comments:

- The abstract and the introduction are well written.

We are grateful for this feedback.

- The following papers can be relevant to the considered research:

1: Nave, O. (2022). A mathematical model for treatment using chemo-immunotherapy. In Heliyon (Vol. 8, Issue 4, p. e09288). Elsevier BV. https://doi.org/10.1016/j.heliyon.2022.e09288

2: Cho, H.-M., Kim, H., Yoo, R., Kim, G., & Kye, B.-H. (2021). Effect of Biofeedback Therapy during Temporary Stoma Period in Rectal Cancer Patients: A Prospective Randomized Trial. In Journal of Clinical Medicine (Vol. 10, Issue 21, p. 5172). MDPI AG. https://doi.org/10.3390/jcm10215172

We appreciated the suggestions of the Reviewer and we added the suggested references in the scoping review in the pertinent sections of the article. Please see lines 35 and 55

- The authors describe the method and immediately after that the result of the paper. The method is not clear. What is the aim of the research? how do the authors achieve their aim? How used the method described?

We adjusted the introduction and the method sections to make the primary aim and the sub-aims clearer to the reader. We reported the PRISMA guidelines of scoping review to outline the research steps that have been followed. Please see attached the references for Scoping Review that have been followed:

- Tricco AC, Lillie E, Zarin W, O’Brien KK, Colquhoun H, Levac D, et al. PRISMA extension for scoping reviews (PRISMA-ScR): Checklist and explanation. Annals of Internal Medicine. 2018.

-Munn Z, Peters MDJ, Stern C, Tufanaru C, McArthur A, Aromataris E. Systematic review or scoping review? Guidance for authors when choosing between a systematic or scoping review approach. BMC Med Res Methodol. 2018;

We hope that after the major revision the whole procedure would be clearer.

- The authors used big data? If yes. Why they didn't use machine learning?

We did not include big data or machine learning analysis because of the nature of the included studies that did not report a large amount of data or quantitative results. Thus, according to the available studies, we opted for a qualitative evaluation of HRV BFB effectiveness on cancer symptoms. 

- The tables present in the paper are not clear. The authors should explain in more detail the present results.

We revised the table format and contents to make them easier to read and more appealing compared to the previous version. We thank the revision for the useful suggestion, we included all methodological and procedural details that were available from the included publications. See tables 1 and 2 page 7-11.

- The conclusion section must be extended. Such that it should include the main results of the paper. The application of the results etc.

We agreed with the suggestion of expanding the conclusions section and we added specific obtained results and clinical implications. Please see the Conclusions paragraph. Please see lines 359-370.

Round 2

Reviewer 3 Report

The authors revised the paper according to my comment. Please edit the paper with native English

Author Response

Dear Editor, we are glad of the positive comments from the Reviewers. As requested, we edited the manuscript with a native English speaker and we re-read the entire manuscript and checked for typos. We hope that this revision helped in improving the quality of the manuscript